# Using community-based participatory approaches to improve access to mass drug administration for trachoma elimination in a pastoral conflict area of Kenya

**Paul M. Gichuki**[1]*, **Bridget W. Kimani**[1], **Tabitha Kanyui**[1], **Collins Okoyo**[1], **Titus Watitu**[2], **Wyckliff P. Omondi**[2], **Doris W. Njomo**[1]

1 Kenya Medical Research Institute (KEMRI), Eastern & Southern Africa Centre of International Parasite Control (ESACIPAC), Nairobi, Kenya, 2 Vector-Borne and Neglected Tropical Diseases Unit (VBNTDU), Ministry of Health, Nairobi, Kenya

* paulmgichuki@gmail.com

**Data Availability Statement:** All relevant data are within the paper.

## Abstract

In Baringo County, Kenya, trachoma remains endemic despite repeated mass drug administration (MDA) efforts, with coverage in one of the wards consistently falling short of world health organization (WHO) targets. The disease is endemic in 12 out of the 47 counties in Kenya. Baringo county is a pastoral conflict, hard to reach area where eight rounds of mass drug administration (MDA) for trachoma have been implemented. In Loyamorok ward, treatment coverage has been below 68% against the WHO recommended threshold of 80%. Community engagements that promote participatory approaches are key to MDA success. In this study, we describe community-based participatory approaches qualitatively developed and implemented during the intervention phase of a study that involved a pre-intervention, intervention and post intervention phases and aimed to address barriers of community participation and access to trachoma MDA. Interviews and focus group discussions were used to identify barriers to community participation in MDA, that included power and gender dynamics, rampant insecurity, community myths and misconceptions, migration in search of water and pastures, vastness and terrain and ineffective teams which resulted to unsupervised swallowing of drugs during MDA campaigns. Stakeholders in trachoma were identified through meetings with national, county and sub-county health management teams. The stakeholders, community members and the research team used the identified barriers to formulate MDA strategies including effective stakeholder engagement, enhanced social mobilization, community awareness creation on trachoma, effective planning and execution of MDA and implementation monitoring of the MDA campaign, all aimed at increasing MDA coverage. Overall MDA coverage in the area increased from 67.6% in 2021 to 87% in 2023 thus meeting the WHO threshold of 80%. The use of community-based, participatory approaches in the development and implementation of data driven strategies has the potential to positively influence MDA coverage for trachoma, and other neglected tropical diseases.

**Funding:** This work received financial support from the United States Agency for International Development (USAID) through its Neglected Tropical Diseases Program through their support of the Coalition for Operational Research on Neglected Tropical Diseases (COR-NTD) grant to DN, through grant number NTDSC 241U. The funders had no role in study design, data collection and analysis, decision to publish, or preparation of the manuscript.

**Competing interests:** The authors have declared that no competing interests exist.

## Author summary

Trachoma is a disease of the eye and a leading infectious cause of blindness in sub-Saharan Africa. It is caused by a bacterium called *Chlamydia trachomatis*. It is transmitted by directly or indirectly transferring of eye and nose discharges from people who already have the disease mostly by flies.

The disease in controlled by among other things provision of drugs to communities where the disease is endemic through mass drug administration (MDA). For the MDA to be effective, more than 80% of the community members should be reached. In areas where communities record very low access to MDA, there is need to identify barriers and develop strategies to mitigate the same. In this study we describe community-based participatory approaches employed to develop and test implementation strategies aimed at increasing access to trachoma MDA in Loyamorok ward of Baringo county. The results showed an increase in community participation in the subsequent MDA, which could be attributed to the implemented strategies.

## Introduction

Trachoma is the leading infectious cause of blindness in sub-Saharan Africa and one of the diseases designated as neglected tropical diseases (NTDs) by the World Health Organization [1]. It is the most common infectious cause of blindness [2]. Blindness from trachoma can be irreversible and the disease is endemic in 44 countries worldwide [3]. Globally, it is responsible for blindness or visual impairment of about 1.9 million people [4]. The disease progresses through repeated infection with conjunctival strains of the bacterium *Chlamydia trachomatis* (*Ct*) resulting in an inflammation (Trachomatous inflammation follicular (TF) and trachomatous inflammation intense (TI) of the conjunctiva known as active trachoma defined within the WHO simplified grading system [5]. Repeated conjunctival inflammation leads to scarring of the eyelid resulting to *trachomatous trichiasis* (TT) that requires surgery as the only treatment [6].

The World Health Organization (WHO) in 1996 recommended the Surgery, Antibiotic, Facial cleanliness and Environmental improvement (SAFE) strategy for the elimination of trachoma as a public health problem globally by the year 2020. Surgery is for *trachomatous trichiasis* (TT), aimed at reducing TT caused by eyelid entropion, administration of antibiotics through mass campaigns to clear infection, facial cleanliness for better personal hygiene and environmental improvement to reduce the risk of infection and reinfection by *Chlamydia trachomatis* [7]. Azithromycin MDA is targeted at all members of the community, with the number of annual MDA rounds depending on the most recent TF prevalence estimate in 1–9 year olds. One round of MDA when TF prevalence is 5.0–9.9%, three MDA rounds when it is 10.0–29.9%, and five rounds when it is ≥30% [8–10]. WHO recommends ≥80.0% treatment coverage with azithromycin MDA to be achieved for trachoma elimination to occur [11].

In Kenya, trachoma was first confirmed as an issue of public health in 2004 [12]. Currently, the disease is endemic in 12 of the 47 counties, including Turkana, Kajiado, Samburu, Laikipia, Marsabit, Isiolo, Kitui, Embu, Meru, Narok, West Pokot and Baringo. Trachoma elimination in Kenya is implemented by the Kenya Trachoma Elimination Programme (KTEP) at the Ministry of Health and aims to eliminate the disease by the year 2027. Under this programme, all members of the affected communities are targeted for treatment during MDA campaign as per the programme dosing guidelines [13].

Implementation of control strategies for trachoma in populations that are not easily accessible such as pastoralist communities who live in conflict prone areas continues to pose a challenge in disease elimination efforts [14]. Pastoralist communities move frequently from their usual communities in search of water and pastures for their livestock, during such times, the communities miss not only on preventive chemotherapy through MDA which greatly affects MDA coverage, but also on other public health interventions [14]. Studies have shown that these communities require public health interventions that are specifically tailored to the nomadic lifestyles [14–16]. Such interventions require a critical understanding of the social, political and economic context of the pastoral communities as studies have shown that the socio-political factors that mostly drive disease transmission in these communities pose challenges in control efforts [17]. Communities living in Loyamorok ward of Baringo county, Kenya are pastoralist and migrate from time to time especially during the dry seasons in search of water and pastures for their livestock. When the migration happens, some members of the families mostly women and children are left behind to take care of their homesteads. During the rainy season, they drive their livestock back and reunite with their families. This area has also experienced conflict resulting from the practice of communities raiding others and stealing their herds of cattle, also referred to as cattle rustling. These factors greatly affect the effectiveness of community based health services such as MDA in the area contributing to lack of achievement of the Sustainable Development Goal (SDGs) pledge to "leave no one behind" [18].

Trachoma is endemic in Tiaty East and West Sub-counties of Baringo county. Control measures against the disease which mainly involved drug administration to the whole community through MDA for trachoma was initiated in the year 2011, with 5 rounds of MDA being implemented up to 2017. This reduced the prevalence of TF from 34.4% to 12.8% [16], prompting administration of a further three more MDA rounds between the year 2020 and 2023 with an aim to reduce the prevalence to the recommended threshold of <5%. Data from the Ministry of Health showed that, Loyamorok ward of Tiaty East sub-county consistently recorded low treatment coverage in all rounds [19]. In all the MDA rounds since 2012, the ward consistently recorded less than 80% coverage. In the 2020 and 2021 campaigns, the coverage was 56.6% and 67.6% respectively. Six out of nine villages in the ward recorded treatment coverage of 48.1%, 48.6%, 49.3%, 55.7% 56% and 57.1% in 2021 [19]. Previous studies have shown that, non-participation in MDA programmes by the affected communities do not occur at random, rather with time, a systematic pattern of non-participation is observed [20], which impacts negatively on the overall MDA coverage [21–24].

Lack of participation in MDA programmes by the target community has been known to hamper successful implementation [23], and is linked to either individual or household level factors [25,26]. Studies have shown the importance of community engagement strategies for successful implementation of MDA programmes for NTDs [27–29]. One such approach is the community-based participatory research (CBPR), an approach that combines knowledge and action to improve health and reduce disparities at the community level [30]. The approach provides a framework to equitably involve community members, researchers and other stakeholders in the research process, thus recognizing and maximizing the importance of each player's contributions and embracing their diversity, towards creating a positive, transformative and sustainable change together [31]. It recognizes communities' strength including the gatekeepers, historical and larger community perspectives, communication styles, and skills thus acknowledging community members as valuable contributors to the process. The researcher team accepts that the communities have their ways of doing things, and such can complement the scientific evidence base [32]. This process of bringing all the stakeholders on the table, also known as co-creation, to develop interventions for issues that affects them has been shown to increase access to health interventions [33,34]. It also serves to increase knowledge and

understanding of a given subject making it easier to integrate it with interventions and social change to improve the health and quality of life of the particular community members [35,36].

Community engagements that promote participatory approaches provide an opportunity for improved awareness creation, community empowerment and ensures that all partners participate in decision making thus facilitating programme ownership by the communities [37]. Therefore, community participatory approach is such a novel idea as it seeks to tap into what the communities know and recognise as their strengths and involve them in the process of problem identification and solution seeking other offering them with solutions as it is with other traditional MDA campaigns. In this paper, we present the strategies developed and tested using community-based participatory approaches to address identified barriers of access to MDA for trachoma in a pastoral conflict area of Kenya.

## Materials and methods

Community engagement through participatory approaches were carried out in the month of May 2023, prior to the launch of the 2023 trachoma MDA in Baringo county which took place on the 20[th] of May 2023. The exercise was aimed at developing and testing implementation strategies which were geared towards increasing community participation and access to MDA in Loyamorok ward. The process involved stakeholders mapping and assembly, validation of barriers and opportunities to trachoma MDA identified earlier by the communities of Loyamorok during the pre-intervention phase of the study conducted between the months of February and March 2023.

### Ethics statement

The study received ethical approval from the Kenya Medical Research Institute—Scientific and Ethics Review Unit (KEMRI/SERU/4532). Further, permission to conduct the study was obtained from Baringo County Health Management Team, local administration and the village elders. Informed consent was obtained from all the stakeholders who participated in the study.

### Study design

The study utilized a pre-intervention, intervention and post-intervention study design. In the pre-intervention phase, data on barriers to participation and access to trachoma MDA were collected using quantitative and qualitative methods. The findings from the pre-intervention phase were then used to formulate MDA Strategy during the intervention phase. The strategy informed actions to improve coverage and effectiveness of trachoma MDA campaign for May 2023 in Loyamorok Ward. The process was a reference for resource allocation, prioritization, and stakeholders' alignment to MDA objectives, and validation of the results during follow up impact evaluation activities. During the post intervention phase, data was collected using both qualitative and quantitative methods to assess the impact of the interventions on the 2023 MDA.

### Study area

Baringo County is located in the former Rift Valley Province. Its headquarters are situated in Kabarnet town which is the largest urban settlement in the county. The County is home to Lake Baringo and has a population of 666,763, and a total of 110,649 households [38]. The County lies between Latitudes 00 degrees 13" South and 1 degree 40" north and Longitudes 35 degrees 36" and 36" degrees 30" East. Its altitude varies between 700m and 3000m above sea level. The county has seven sub-counties including Tiaty East, and Loyamorok ward where the study was carried out is one of the four wards in Tiaty East sub-county. The ward has a

population of 13,885 and covers an area of 597.8km$^2$. The natives are pastoralists who migrates during the dry seasons in search of water and pastures for their livestock but leave some of their family members behind. When it rains, they drive their livestock back and reunite with their families. The area is also characterized by conflicts resulting from acts of cattle rustling, which traditionally has been considered as a cultural practice. Their main language of communication is *Pokot* and the main economic activity is animal trading, that includes cows, goats, sheep, donkeys, and camels which happens mostly during designated market days [38].

## Steps to MDA strategy formulation

During the intervention phase of this study, the findings from the pre-intervention phase were used to formulate strategies to inform actions to improve coverage and effectiveness of the MDA Campaign for May 2023 using community participatory approaches. The following stages were followed.

### Stage 1: Stakeholder's mapping

The research team held planning meetings with the national trachoma coordinator, Baringo county health management team and the Tiaty East sub-county health management team to identify key players in trachoma MDA campaign in the study area. The team identified various stakeholders owing to their previous roles in trachoma work in the area, and their influence in the community (Table 1).

### Stage 2: Participatory design of strategies

The research team worked closely with the national, county and sub-county NTD focal persons and key community representatives including the chiefs and *wazee wa nyumba kumi* (village elders) to co-plan the implementation of a multi sectoral stakeholder's meeting. A two days' workshop was called for all the identified stakeholders, community representatives and the research team. During the workshop, an introduction of the team was done where people paired up with the person sitting next to them, and each learnt at least five things about them. Each person was then given an opportunity to introduce their partner. This ensured that every person participated, learnt and expressed themselves, as a way of creating cohesion among the team. Each group represented in the meeting was given a role to play during the workshop, which served to encourage more participation and also build trust. The meeting was co-facilitated by members of the research team, the national, county and sub-county NTD focal personnel for ownership and government leadership in developing interventions to respond to gaps identified during the study.

The research team presented on barriers to community participation in MDA which had been identified earlier by the communities during the pre-intervention phase of the study. After this, multi-disciplinary teams were formed for group work and reporting back to the plenary where the individuals brainstormed potential strategies, discussed ideas and experiences which ensured divergent views on key challenges and possible solutions were captured for considerations. Each group were facilitated with record cards to capture their feedback and inputs. After the group discussion sessions, the whole team converged for plenary discussions on the group presentations and worked towards converging and building consensus on the key challenges and possible, doable solutions [39]. All the emerging issues were captured on displayed flip charts where everyone could see. This was followed with an iterated generation of strategies, interventions and activities while answering the questions, Why? What? How? and When?. This process was aimed at having the teams identify gaps in the planned subsequent MDA. At the end of the engagement, a list of actionable tasks and their costs were proposed for consideration and implementation consistent with the 2023 MDA schedule. This ensured

**Table 1. Stakeholders who attended the strategy workshop.**

| Stakeholder | Gender | Organization |
|---|---|---|
| National trachoma focal person | M | MOH |
| County NTD coordinator | M | Baringo county |
| County Trachoma Focal Person | M | Baringo county |
| County Community Health Focal Person | F | Baringo County |
| County Public Health Officer | M | Baringo County |
| Specialist Community Public Health Nurse | F | Baringo County |
| Ministry of Education (MOE) | F | Tiaty East Sub -County |
| Ministry of Water, Sanitation and Irrigation | M | Tiaty East Sub -County |
| Business person | F | Tiaty East Sub -County |
| Health Records Information Officer (HRIO) | M | Tiaty East Sub -County |
| Sub county Disease Surveillance Coordinator | M | Tiaty East Sub -County |
| Sub County Eye Health | M | Tiaty East Sub -County |
| Community Health Worker | F | Tiaty East Sub -County |
| Assistant County Commissioner (ACC) | M | Tiaty East Sub -County |
| Health Records Information Officer (HRIO) | F | Tiaty East Sub -County |
| Ward Administrator | M | Loyamorok Ward |
| Chief | F | Loyamorok Ward |
| Sub chief | M | Loyamorok Ward |
| Community member | M | Loyamorok ward |
| Nurse | F | Loyamorok Ward |
| Community Based Organization (CBO) member | F | Loyamorok Ward |
| Community Based Organization (CBO) member | M | Loyamorok Ward |
| Youth Leader | M | Loyamorok Ward |
| Women group chairlady | F | Loyamorok Ward |
| Religious leader | M | Loyamorok Ward |
| Fred Hollows Foundation | F | NGO |
| Total participants | | 26 |

that the developed strategies for implementation were based on the experiences of community members and were feasible for implementation. The stakeholders, community and the research team then discussed each actionable task and the feasibility and practicability of implementation settling for the most feasible and doable actions. This process led to prioritization of the key strategies, interventions and activities. The communities were actively involved in the delivery of the interventions during the MDA as described in the results section.

## Trachoma MDA coverage estimation

The MDA coverage data was estimated using administrative data, where the numerator is the number of persons treated as recorded by drug distributors during MDAs for the numerator, and the existing population estimates usually from the national census forms the denominator. During the MDA, a supervision coverage tool (SCT) was used as a monitoring tool to assess the coverage and quality of drug distribution process [40].

## Results

### Barriers identified and validated during stakeholder's engagement forum

Several barriers to community engagement and access to trachoma MDA were identified during the pre-intervention phase including the following.

**Power and gender dynamics.**   Communities living in the study area culturally regards the place of women and children lower than that of men. The place of women in the community was lower than that of men and in such there were cultural practices to be considered example, men did not share cups with women and children as such administration of the drugs would require careful planning. It was proposed that the MDA teams consider reaching the men at different times respecting the societal norms. In the community, women were reported to have specific times they would be available at home, market places, and water points. These were considered in the development of daily movement plans.

**Rampant insecurity in the area.**   There were reported insecurity incidences in the area due to banditry and cattle rustling. It was reported that counter measures had improved weeks to the planned MDA campaign although the risk of deterioration remained high and mitigation planning was pertinent. It was suggested that engagement of the security managers at county and local levels was key to the success of the MDA campaign.

**Community myths and misconceptions.**   Results from the pre-intervention phase of the study showed that community members have inadequate knowledge about trachoma and those who reported to be knowing the disease believed that it was caused by dust, dirty water, inheritance and old age. The causes of trachoma disease were also not known. There was also the belief among men that the drugs given for trachoma will affect their libido and also cause stomach problems. The stakeholders suggested the need for community awareness creation to close trachoma knowledge gap.

**Prolonged drought leading to migration in search of pastures.**   It was reported that, the area often experiences extended periods of drought. During such periods, significant number of community members migrate with their animals in search of pastures, but leaves some members of their families behind. When MDA happen during such periods, most of the community members who have migrated are missed. The stakeholders suggested that the next MDA be planned to take place immediately after the rains when the communities have settled down.

**Accessibility challenges due to vastness of terrain.**   Loyamorok ward is expansive, thus some areas are far and with poor accessibility. There were reported cases of wild animals and poisonous snakes which may limit people's movement during MDA. It was agreed that motorbikes would be a better means of accessing these areas so as to maintain timely and constant supplies of the MDA drugs. It was observed that some community members might not be reached due to old age or illness, and thus it was suggested that home visits to be carried out using motorbikes to administer the drugs.

**Ineffective distribution of teams leading to unsupervised swallowing of drugs.**   On this, it was reported that it could have been due to collecting of MDA drugs for persons at home or husbands who avoided the distribution points or sick and elderly persons. To enhance effectiveness of the campaign, teams were discouraged from issuing drugs unless they could verify the swallowing. The team finally proposed that MDA campaign teams be drawn from local people who fully understood the targeted areas, and who were motivated to meet the planned targets for the MDA campaign. The planning team was encouraged to engage persons based on community referrals that had proven record to deliver on assigned tasks. There were concerns that teams selected on any other than merit including knowledge of the target areas would not be fully motivated and engaged to meet the planned targets for the MDA Campaign.

## Key strategies and interventions designed and implemented

The stakeholder's engagement workshop prioritized five key strategies for anchoring interventions during the 2023 MDA campaign for Loyamorok (summarized in Table 2). They included, effective stakeholder coordination and engagement, community awareness creation

**Table 2. Key strategies and interventions.**

| Identified barrier | Description | Key strategy/ strategies | Interventions |
|---|---|---|---|
| Power and gender dynamics | The place of women and children was considered lower than that of men in the community | Effective stakeholder's coordination and engagement | -Establishing the different stakeholders in the community<br>-Establishing specific roles of each stakeholder<br>-Defining their engangement at the different stages of MDA campaign including planning, actual campaigns and post MDA outcome monitoring |
| Rampant insecurity | Rampant insecurity and conflict attributed to cattle rustling | Effective stakeholder's coordination and engagement | -Engagement of the security apparatus in the area<br>- Multi-sectoral stakeholder meetings<br>- Security team meetings led by ACC, chiefs, and assistant chiefs' *barazas*<br>-Involvement of the security apparatus during the MDA campaigns |
| Myths and misconceptions about trachoma. | Inadequate knowledge about trachoma disease which results to myths and misconceptions about the disease | Community awareness creation to close trachoma knowledge gap | -Effective communication<br>-Right messaging<br>-Using opinion leaders, MDA champions, and gate keepers to communicate key messages through local channels |
| Migration due to prolonged drought | People migrate in search of water and pastures for their livestock | Effective planning and execution of the MDA campaign | -Co-planning of the intended activities with the affected community.<br>- Engagement of stakeholders in designing the activity implementation plan was prioritized<br>- Timing of MDA to happen after the rains |
| Accessibility challenge | -The vast terrain in the area poses accessibility challenge<br>-Limited time to conduct MDA<br>-Lack of MDA progress tracking | Activity Implementation Monitoring | -Daily Movement Plans<br>-Daily Activity Returns<br>-Peer to Peer WhatsApp Groups<br>-Feedback and support meetings at Ward level<br>-Motor cycles availed<br>-County level supervision<br>-Supervised drug swallowing |
| Ineffective teams leading to unsupervised swallowing of drugs | -Irregular allocation of personnel to support the MDA campaign.<br>-Low motivation of the MDA teams for full engagement<br>- | Effective Planning and Execution of MDA Campaign | -Enhanced capacity of the MDA Campaign teams<br>-Context-informed MDA Campaign implementation<br>-Mapping and right allocation of teams to effectively cover the MDA sites |

to close trachoma knowledge gap, effective planning and execution of MDA campaign, enhanced social mobilization for increased coverage and activity implementation monitoring.

**Effective stakeholder coordination and engagement.** This strategy was aimed at addressing the barriers related to power and gender dynamics as well as insecurity during the MDA period. Specific roles of each stakeholder and how they were to be engaged at the different stages of MDA campaign including planning, actual campaigns and post MDA outcome monitoring. The strategy expounded the objective of stakeholder engagement, the key interventions, the players involved, the methods and it also outlined the responsibilities of each. The key guiding principles included inclusivity, transparency, appropriateness, clarity, comprehensiveness and MDA timing so as to happen when people have not migrated in search of pastures. The interventions agreed upon in this strategy included multi sectoral stakeholder's meetings at the ward level and regular security meetings headed by the Assistant county commissioners (ACC), chiefs and the sub chiefs (Table 2).

**Community awareness creation to close trachoma knowledge gap.** In this strategy, the stakeholders aimed to educate the community about trachoma disease with a focus to influencing their attitudes, behaviour and beliefs towards participating in the MDA campaign. Messages were tailored to the local context and intensively delivered to the communities through local platforms. (Table 2).

**Effective planning and execution of MDA campaign.** This strategy was geared towards closing the barrier of community migration so that all the members of the community get a chance to participate in the forthcoming MDA. The stakeholders agreed that co-planning with the beneficiary community was key in achieving the MDA goals. Study findings were reviewed together and areas that required strengthening to make a difference in the MDA were identified. Allocation of additional resources to facilitate effective reach to the inaccessible populations in the Study area. (Table 2).

**Enhanced social mobilization for increased coverage.** This strategy was aimed at addressing the challenge of accessibility to drugs due to vastness of the terrain in the area. It was noted that timely engagement of communities, and having them understand the benefits of participating in the MDA was key. During the participatory engagement, some key avenues identified by stakeholders included empowering the local MOH officials to be agents in encouraging the local communities to come out in large numbers and participate in MDA for trachoma. The MOH officials were also tasked to participate in meetings and events by local communities and proactively design, organize and implement activities while appreciating the culture of the target community. The stakeholders adopted the Social Behaviour Change and Communication (SBCC) Strategy to promote changes in knowledge, attitudes, norms, beliefs and behaviours related to trachoma. The stakeholders' meetings identified sub-populations and community groups that were reached with relevant messages about trachoma and planned MDA Campaign. Different channels for tailor-made communication were proposed for use pre and during the MDA campaign. These included the use of mass media (radio), youth and women groups' communication structures, the local administration platforms, schools, churches, and market places. Posters and public address (PA) systems were proposed as tools of communication (Table 2).

**Activity implementation monitoring.** To ensure that there was a balanced team formation and deployment during the MDA implementation, the recommendations of the stakeholders was strictly adhered to during recruitment and team composition. Training of the teams where the issues of unsupervised swallowing of drug was addressed. During the actual MDA, progress was monitored closely with daily reports being made to the supervisors to ensure achievement of expected results, spotting bottlenecks in implementation and highlighting any unintended effects (positive or negative) that needed relooking. During discussions with the stakeholders on the study findings and design of interventions, the need for strengthening and where necessary putting in place mechanisms for monitoring the implementation of the MDA Campaign 2023 was discussed. Supervision for swallowing of the azithromycin drug given during the MDA was reported as an area that required full enforcement to ensure effectiveness on the campaign. Daily monitoring of the coverage returns at ward level through reports, meetings, and WhatsApp groups were some mechanisms highlighted for use. The Study team was to be roped with these mechanisms for full engagement (Table 2).

## Cost implication

The community based participatory approaches implemented in this study came with additional costs. Costs associated with live local radio sessions and recorded messages sessions urging communities to take part in the MDA were played frequently before and during MDA, a public health system with a courier was engaged during the market days, and two additional motorbikes were provided for supervision during the MDA. These costs were covered by this study.

## May 2023 trachoma MDA coverage in Loyamorok ward

The overall MDA coverage in the area increased from 67.6% in the previous MDA to 87% during the 2023 MDA thus meeting the WHO threshold of at least 80%. The MDA coverage data

was estimated using administrative data, where the numerator is the number of persons treated as recorded by drug distributors during MDAs for the numerator, and the existing population estimates usually from the national census forms the denominator. During the MDA, a supervision coverage tool (SCT) was used as a monitoring tool to assess the coverage and quality of drug distribution process [40].

## Discussion

In this study, we used community based participatory approaches to design strategies for improving access and participation in MDA for trachoma elimination. Co-creation approaches where all the stakeholders are involved in the process of searching for solutions to issues affecting communities not only helps the affected populations to own the process but also gives them a voice [33,34,41]. The approach of stakeholder's validating what the community reported as barriers to trachoma MDA was key for acceptability and ownership, thus paving way for them to work together in coming up with strategies of overcoming the barriers.

Previous studies have opined that for community engagement strategies to be effective in facilitating community participation, there must be adequate partnership approaches which seeks to address health issue and should address context specific implementation barriers affecting community participation in MDA [42]. Community participatory approaches explored and came up with different strategies, pointing at local feasible solutions to address the identified barriers to community participation and access to MDA.

### Stakeholder's engagement

This strategy was aimed at addressing the barriers related to power and gender dynamics as well as migration of the communities which is occasioned by prolonged drought, conflicts due to cattle rustling and insecurity during the MDA period. Involvement of the key stakeholders, both from the health sector, security and the local leadership at the community level in decision making was key in ensuring support for the MDA campaign by the local population. Establishing specific roles of each stakeholder and clearly outlining their responsibilities at the same time sharing common guiding principles ensured smooth cascading of the right information to the community members. Having previously experienced sustained conflict and insecurities resulting mainly from cattle rustling, the stakeholder's engagement ensured a coordinated multi sectoral stakeholder's meetings from the lowest level and regular security meetings headed by the Assistant County Commissioners (ACCs), chiefs and the sub chiefs who came up with elaborate plans to beef up security during the MDA period. Involvement of the security apparatus right from the planning to the execution phase of the MDA was critical in this study. Our findings affirm that involvement of the local community leadership structures in MDA activities is essential for maximum community participation [2,43,44].

### Social mobilization

Social mobilization was used as a tool to address the barriers of community myths and misconceptions about trachoma disease. Strategies involving enhanced social mobilization and community awareness creation for increased coverage of MDA was implemented both prior to and during the MDA campaign. The stakeholders' meetings identified sub-populations and community groups with potential to influence other community members and targeted them with relevant messages about trachoma. These included women groups, public meetings (*barazas*), religious and social platforms (schools, churches, markets), and places of men in the community (*kokwo).* Different channels for tailor-made communication were proposed for use pre and during the MDA campaign. These included the use of mass media (radio), youth

and women groups' communication structures, the local administration platforms, schools, churches, and market places. Posters and public address (PA) systems were proposed as tools of communication. Empowering community members with the right information motivated them to take part in the subsequent MDA. Social mobilization at community level has been reported as a key driver in having communities own the process [45,46] and that working across networks serves to increase information uptake by the communities [47,48] which leads to success of the activities being implemented. In the current study, available social structures were used in social mobilization supporting previous findings which have shown importance of local social structures. [49,50].

### Addressing pastoral migration

The study was carried out in a pastoralist community where members of the community migrate during the dry seasons in search of water and pastures for their livestock. When it rains, they drive their livestock back and reunite with their families. This therefore required effective planning and execution of the subsequent MDA as a strategy to overcome the barrier. It was noted that previous trachoma MDAs in the study area had taken place mostly during the dry seasons, when most people in the community were likely to have migrated to other areas and therefore missing out. Previous studies have reported migration among the pastoralist communities as an important barrier to the success of MDA [14,51]. In the present study, MDA was planned and implemented during the rainy season when there was not much movement by the communities since there was enough water and pastures for the animals. Being a rainy season, the weather was also favourable to the implementation teams who face challenges walking in very hot weather.

### Logistical planning

During the stakeholder's planning meetings, MDA campaign areas were mapped out and daily movement plans for drug distributers, community mobilizers and the supervision team were developed putting into consideration the terrain and high population target areas. Areas which were harder to reach were allocated more staff, and the access routes availed through the administrative office. Motor cycles were also availed for use. Previous studies have alluded to the critical role that geographical factors such as vastness and terrain, and long distance to be covered during the MDA plays in hindering success of health campaigns including MDA [16,52–54]. Adequate teams were recruited putting into consideration their qualifications, knowledge about the study area and previous experience in trachoma work in accordance to the MDA activity implementation and monitoring strategy. Resources were allocated to the teams for use in the field including airtime to facilitate communication and motorbikes. The road network in the area was a challenge as there were some areas where there were no roads at all. In such places the teams had to cover long distances. Throughout the process, the security teams were easily accessible which gave the communities confidence. Coordinated movement of the teams was ensured with daily activity returns to the supervising teams at the ward level. There was a team composed of members of CHMT and SCHMT carrying out field support supervision. The teams also held daily meetings to discuss on planned targets and achieved results for Loyamorok ward. This approach ensured that the MDA program was tailored to the local condition by the local population thus creating ownership and collective community responsibility as described in previous studies [24,55,56].

### Study limitations

A limitation of this study is that majority of stakeholders involved were from Tiaty East subcounty, and that study might also have missed on some sub-groups in the community. It is

possible that the study would have benefited from more insights and contributions from stakeholders from other sub-counties. The study used previous coverage data as a baseline and introduced some interventions and again assessed the changes in coverage without a comparison group and therefore the changes observed may not with certainty be associated with the interventions, as there could be other overriding factors. Community based participatory approaches are more intensive than the traditional planning methods and requires additional resources which may affect its sustainability.

## Conclusion

This study found out that use of community based, participatory approach in the development and implementation of data driven strategies has the potential to positively influence trachoma MDA coverage in an area that had consistently recorded low uptake. This suggests that the participatory approach is key in involving communities and stakeholders in decision making process for community based issues, especially for communities living in hard-to reach, nomadic, conflict prone areas. It is also clear from the study that it is within the community's power to identify reasons which make them ward off from participating in MDA, and it's also within their domain to suggest local based solutions which are doable and achievable. Future public health interventions, particularly those targeting neglected tropical diseases and community-driven health issues, should integrate participatory approaches to ensure long- term success and sustainability.

## Acknowledgments

We wish to acknowledge the immense support accorded to the study by the National trachoma coordination team, Baringo county health leadership, Tiaty East sub county health management team, all stakeholders for their technical support during the implementation of this study. We are also grateful to the people of Loyamorok for their participation and patience during data collection activities. This study has been published with the permission of the Director General, KEMRI.

## Author Contributions

**Conceptualization:** Doris W. Njomo.

**Data curation:** Paul M. Gichuki, Collins Okoyo.

**Formal analysis:** Paul M. Gichuki.

**Funding acquisition:** Doris W. Njomo.

**Investigation:** Paul M. Gichuki, Bridget W. Kimani, Tabitha Kanyui, Collins Okoyo, Titus Watitu, Wyckliff P. Omondi, Doris W. Njomo.

**Methodology:** Paul M. Gichuki, Bridget W. Kimani, Collins Okoyo, Titus Watitu, Doris W. Njomo.

**Project administration:** Paul M. Gichuki, Titus Watitu, Wyckliff P. Omondi, Doris W. Njomo.

**Resources:** Doris W. Njomo.

**Supervision:** Paul M. Gichuki, Bridget W. Kimani, Tabitha Kanyui, Titus Watitu, Wyckliff P. Omondi.

**Validation:** Collins Okoyo, Doris W. Njomo.

**Writing – original draft:** Paul M. Gichuki.

**Writing – review & editing:** Paul M. Gichuki, Bridget W. Kimani, Tabitha Kanyui, Collins Okoyo, Titus Watitu, Wyckliff P. Omondi, Doris W. Njomo.

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
