## [Decision Letter · Decision Letter 0]

1 Oct 2024

Dear Dr Gichuki,

Thank you very much for submitting your manuscript "Using Community-Based Participatory Approaches to Improve Access to Mass Drug Administration for Trachoma Elimination in a Pastoral Conflict Area of Kenya" for consideration at PLOS Neglected Tropical Diseases. As with all papers reviewed by the journal, your manuscript was reviewed by members of the editorial board and by several independent reviewers. The reviewers appreciated the attention to an important topic. Based on the reviews, we are likely to accept this manuscript for publication, providing that you modify the manuscript according to the review recommendations. 

Sincerely,

Elsio A Wunder Jr, DVM, Ph.D.

Section Editor

Elsio Wunder Jr

Section Editor

Reviewer's Responses to Questions

**Key Review Criteria Required for Acceptance?**

**Methods**

-Are the objectives of the study clearly articulated with a clear testable hypothesis stated?

-Is the study design appropriate to address the stated objectives?

-Is the population clearly described and appropriate for the hypothesis being tested?

-Is the sample size sufficient to ensure adequate power to address the hypothesis being tested?

-Were correct statistical analysis used to support conclusions?

-Are there concerns about ethical or regulatory requirements being met?

Reviewer #1: (No Response)

Reviewer #2: (No Response)

**Results**

-Does the analysis presented match the analysis plan?

-Are the results clearly and completely presented?

-Are the figures (Tables, Images) of sufficient quality for clarity?

Reviewer #1: (No Response)

Reviewer #2: (No Response)

**Conclusions**

-Are the conclusions supported by the data presented?

-Are the limitations of analysis clearly described?

-Do the authors discuss how these data can be helpful to advance our understanding of the topic under study?

-Is public health relevance addressed?

Reviewer #1: (No Response)

Reviewer #2: (No Response)

**Editorial and Data Presentation Modifications?**

Reviewer #1: (No Response)

Reviewer #2: (No Response)

**Summary and General Comments**

Reviewer #1: (No Response)

Reviewer #2: Dear Editorial Team / Authors,

I have reviewed the manuscript titled “Using Community-Based Participatory Approaches to Improve Access to Mass Drug Administration for Trachoma Elimination in a Pastoral Conflict Area of Kenya” (Manuscript Number: PNTD-D-24-00973). The study addresses an important public health issue, but there are several areas that could benefit from revisions to improve clarity, coherence, and scientific rigor. Below are my detailed comments and recommendations:

1. Abstract:

1.1. Clarity and Focus (Line 1): The first sentence, “Trachoma, a neglected tropical disease (NTD) is the leading infectious cause of blindness in sub-Saharan Africa,” is quite general and familiar to the journal’s readership. I suggest rephrasing it to focus more on the study’s context. For instance: “In Baringo County, Kenya, trachoma remains endemic despite repeated mass drug administration (MDA) efforts, with coverage consistently falling short of WHO targets.”

1.2. Methods Section: The methods section of the abstract could be more explicit. Please specify the participatory approaches employed—was it qualitative? Were focus groups or surveys utilized? Briefly outline how stakeholders were identified and involved.

1.3. Results Section: While the results are well-stated, the phrase *“community-based participatory approaches significantly increased trachoma MDA coverage”* is somewhat vague. If space allows, provide additional details on the specific interventions or changes that contributed to this improvement.

1.4. Conclusion: The conclusion is clear but would be strengthened by a sentence discussing the broader implications for future MDA campaigns or for other neglected tropical diseases (NTDs).

2. Introduction:

2.1. Clarifying the Knowledge Gap: While the introduction provides a broad overview of trachoma and its control, it would benefit from clearly articulating the knowledge gap this study addresses. What makes the participatory approach novel in this context compared to traditional MDA campaigns? Explicitly stating this would strengthen the study’s rationale.

2.2. Emphasizing Participatory Approaches: The introduction could introduce community-based participatory research (CBPR) earlier and explain why it is particularly suited for this context. Additionally, highlighting how CBPR improves upon previous MDA strategies would better justify the need for this study.

2.3. Expand on Pastoralist Dynamics: The section on pastoralist migration patterns is essential for understanding the challenges of MDA delivery. Consider expanding this by discussing how migration affects public health interventions, potentially referencing similar studies on health delivery in migratory communities.

3. Discussion:

3.1. Improve Flow and Coherence: The discussion moves between various topics (stakeholder engagement, social mobilization, migration) without a clear structure. I recommend organizing the section into distinct sub-sections, such as “Stakeholder Engagement,” “Social Mobilization,” “Logistical Planning,” and “Addressing Pastoral Migration.”

3.2. Repetition of Concepts: Some points, especially those related to social mobilization and communication strategies, are repeated. Consolidating these discussions will make the manuscript more concise and focused.

3.3. Balancing Strengths and Challenges: While the discussion highlights the successes of the intervention, a more balanced perspective that acknowledges challenges and unexpected difficulties would strengthen the scientific rigor of the paper.

3.4. Expand on the Role of Security and Migration: The involvement of security forces and the timing of the MDA campaign during the rainy season are presented as key strategies, but further explanation is needed. Did security involvement build community trust? Were there any resistance or challenges related to security? How did participation in the rainy season compare to past campaigns?

4. Limitations:

4.1. Broaden the Scope of Limitations: The limitation mentioned is valid but could be expanded to address additional points:

4.2. Selection Bias in Stakeholder Engagement: Were all community sub-groups adequately represented in the participatory process? If not, how might this have affected the outcomes?

4.3. Generalizability: Given that this study focuses on a specific pastoralist community, discuss how transferable the findings are to other regions or contexts, particularly those with different migration or conflict dynamics.

4.4. Unintended Consequences: Were there any unintended negative outcomes from involving security forces, or other elements of the strategy? For instance, did the presence of security create tension or fear within certain population groups?

4.5. Long-Term Sustainability: Address the sustainability of these participatory approaches. Will the community maintain high levels of participation in future MDAs without continued intensive engagement?

5. Conclusion: The last sentence, “Community-based participatory approaches should be adopted for other health interventions,” is somewhat broad. I suggest rephrasing it to be more specific: “Future public health interventions, particularly those targeting neglected tropical diseases and community-driven health issues, should integrate participatory approaches to ensure long-term success and sustainability.”

Best regards, 

Reviewer

PLOS authors have the option to publish the peer review history of their article (what does this mean?). If published, this will include your full peer review and any attached files.

Reviewer #1: Yes: Margaret Baker

Reviewer #2: Yes: Hamidreza Hasani

Figure Files:

Data Requirements:

Reproducibility:

References

---

## [Editor Report · Decision Letter 1]

24 Oct 2024

Dear Dr Gichuki,

We are pleased to inform you that your manuscript 'Using Community-Based Participatory Approaches to Improve Access to Mass Drug Administration for Trachoma Elimination in a Pastoral Conflict Area of Kenya' has been provisionally accepted for publication in PLOS Neglected Tropical Diseases.

Best regards,

Elsio A Wunder Jr, DVM, Ph.D.

Section Editor

Shaden Kamhawi

co-Editor-in-Chief

Paul Brindley

co-Editor-in-Chief

---

## [Editor Report · Acceptance letter]

29 Oct 2024

Dear Dr Gichuki,

We are delighted to inform you that your manuscript, "Using Community-Based Participatory Approaches to Improve Access to Mass Drug Administration for Trachoma Elimination in a Pastoral Conflict Area of Kenya," has been formally accepted for publication in PLOS Neglected Tropical Diseases.

Best regards,

Shaden Kamhawi

co-Editor-in-Chief

Paul Brindley

co-Editor-in-Chief
